# ON SAMPLING INFORMATION SETS TO LEARN FROM IMPERFECT INFORMATION

## ABSTRACT

In many real-world decision-making scenarios, agents are confronted with incomplete and imperfect information, requiring them to make choices based on limited knowledge. Imperfect-information games tackle this challenge by organising different potential situations into so-called information sets, i.e. sets of possible world states that are indistinguishable from one observer's perspective, but directly evaluating an information set is difficult. A common but often suboptimal strategy is to evaluate the individual states in the set with a perfect information evaluator and combine the results. This not only presents problems related to translating perfect information evaluations to imperfect information settings but is also immensely costly in situations with extensive hidden information. This work focuses on learning direct evaluators for information sets by assessing only a subset of the states in the information set, thereby reducing the overall cost of evaluation. Critically, we focus on one question: How many states should be sampled from a given information set? This involves a trade-off between the cost of computing a training signal and its accuracy. We present experimental results in three settings: an artificial *MNIST* variant with hidden information, *Heads-Up Poker*, and *Reconnaissance Blind Chess*. Our results show that the number of sampled states significantly influences the efficiency of training neural networks. However, there are diminishing returns when sampling a large number of states. Notably, in the three regarded domains, using one, two and two samples respectively leads to the best performance concerning the total number of evaluations required. This research contributes to the understanding of how to optimise the sampling of information sets in scenarios of incomplete information, thus offering practical insight into the balance between computational cost and accuracy.

## 1 INTRODUCTION

Imperfect-information games, games characterised by unobservable aspects, are an important part of Game AI research. In recent years, they have received increased attention due to the inherent complexity of managing incomplete information. This category encompasses a wide array of games, spanning from classical card games like Poker and Bridge to adaptions of traditional board games such as Dark Hex and Reconnaissance Blind Chess, as well as real-time video games like Starcraft, Dota II and Counter-Strike. Thus, we see much interest - commercially and scientifically - in mastering this category of games. However, the methods that conquered many classical perfect-information games like AlphaZero (Silver et al., 2018) do not necessarily carry over to imperfect-information games (Schmid et al., 2021) easily and mostly require specialised techniques.

In imperfect-information settings, decisions are typically based on a fusion of public information and an implicitly learned or directly computed expected value of the hidden information. While there are several different approaches to learning evaluations implicitly (see Section 3), we focus on learning them explicitly in a supervised fashion. Our central concept revolves around receiving training signals for imperfect information states via the expected value of all possible perfect information states. At every decision point of an imperfect-information game, the set of all possible states from one observer's perspective is called an *information set*. While enumerating such a set may not be possible for real-time games, it is feasible for many sequential games such as *Poker* and *Reconnaissance Blind Chess*. We define the evaluation of an imperfect-information state as the expected value over all evaluations of states in its information set. The goal is to learn an

evaluator which encapsulates this relationship between and imperfect-information state and a target evaluation. However, doing this perfectly would require evaluating every state across all obtainable information sets, which is often not a feasible task. Thus, this work investigates how we can reduce the computational work required by only sampling subsets of each information set.

We begin this work by outlining the problem more formally (Section 2) and give a brief overview of related work and problems (Section 3). Subsequently, we empirically investigate the problem in three settings with different types of hidden information:

- **MNIST with uncertainty**: This introduces the general concept by corrupting the training labels with which a classifier is trained.
- **Heads-Up Poker**: Here we evaluate a 2-card hand without knowledge about the community cards or the opponent's cards, sampling from information sets to estimate evaluations.
- **Reconnaissance Blind Chess (RBC)** (Gardner et al., 2019): In this chess variant, the opponent's moves are often uncertain because of limited information. We aim to evaluate the public state based on evaluations of determinised positions.

We summarise our results in Section 5 and give an outlook on potential future extensions in Section 6.

## 2 PROBLEM STATEMENT

We formalise the problem as follows: Given is a dataset of examples $\mathcal{D} = (\mathbf{x}_i, y_i) \subset X \times Y$, where each label $y_i = f(\mathbf{x}_i, \mathbf{h}_i)$ is determined by an unknown function $f$, dependent not only on the observable information $\mathbf{x}_i$, but also on the hidden information $\mathbf{h}_i$. Our goal is to find a function $g(\mathbf{x})$ which approximates $f(\mathbf{x}, \mathbf{h})$, such that $\forall i : g(\mathbf{x}_i) \approx f(\mathbf{x}_i, \mathbf{h}_i)$. Obviously, this task is non-trivial, and such a function $g$ does not always exist, as the same observable $\tilde{\mathbf{x}}$ can occur multiple times with different labels because in general $f(\tilde{\mathbf{x}}, \mathbf{h}^{(1)}) \neq f(\tilde{\mathbf{x}}, \mathbf{h}^{(2)})$ for $\mathbf{h}^{(1)} \neq \mathbf{h}^{(2)}$.

Our motivation for this problem originates from imperfect information games, where the information set represents all possible game states given one player's information. In several such games, remarkable performance has been achieved by basing the imperfect information gameplay, whether implicitly or explicitly, on perfect-information evaluations of states in an information set (Blüml et al., 2023; Bertram et al., 2022; Browne et al., 2012). In *RBC*, many strong programs rely heavily on classical engines for evaluating conventional chess positions (Gardner et al., 2019; Perrotta et al., 2021; Gardner et al., 2023). The idea is to evaluate the public information state by the expected value of the states in the information set. Similarly, the value of a player's hand in *Poker* can be estimated as the expected value of the hand over all possible variations of the community and opponent's cards.

It is important to acknowledge the limitations of basing imperfect-information policy fully on perfect-information evaluations, and it is trivial to construct counterexamples where this fails. Nevertheless, often no better estimates exist and learned evaluations can subsequently be refined through reinforcement learning or other techniques.

The central objective of this work is to learn the function $g$ which receives the public information of a state $\mathbf{x}$ and approximates the expected value of that state. This expectation is received by iterating over the information set:

$$\hat{y} = \sum_{\mathbf{h}} P(\mathbf{h}|\mathbf{x}) \cdot f(\mathbf{x}, \mathbf{h}) \tag{1}$$

Here, $\mathbf{h} \in \mathcal{I}$ are all possible configurations of private information that are part of the information set $\mathcal{I}$, $f$ is an evaluator of a perfect information state and $P$ is a function which gives the probability of each hidden state for the given configuration $\mathbf{x}$. In our experiments, we assume that all possible determinations are equally likely, i.e. $P(\mathbf{h}|\mathbf{x}) = 1/|\mathcal{I}(\mathbf{x})|$. In general, $P$ can be more complex and can be heuristically approximated based on past behaviours or observations (Bertram et al., 2023).

A simple strategy to learn $g$ is to collect samples of the form $(\mathbf{x}_i, \hat{y}_i)$, i.e., to compute the exact value $\hat{y}_i$ as in equation 1 for a large number of training positions $\mathbf{x}_i$, and to use supervised learning to

learn the function $\hat{y}_i = g(\mathbf{x}_i)$ from these samples. However, this approach generally is too costly due to the potentially large size of information sets, so obtaining a single $\hat{y}_i$ would require tens of thousands of queries to the evaluator. Alternatively, $\hat{y}$ can be approximated by randomly sampling only a few of the possible $\mathbf{h}^{(j)}$, resulting in less accurate training signals $\dot{y}$ at a lower computational cost.

In our work, we aim to answer a fundamental question: Given a fixed budget of $N$ perfect information evaluations, how should we generate training data for the learner? Options range from generating $N$ different training examples $\mathbf{x}_i$, each labelled with one randomly sampled evaluation, over using a fixed number of $k$ evaluations to generate labels for $N/k$ positions, up to exhausting the budget with exactly computing $\hat{y}$ for as many examples as possible. This trade-off between the training set size (the number of distinct $\mathbf{x}_i$) and label quality (the number of evaluations used to estimate the intended target values $\hat{y}_i$ for each $\mathbf{x}_i$) forms the core focus of this paper.

Several learning settings are special cases of this formulation. Conventional classification emerges when $\mathbf{h}_i = \emptyset, \forall i$, i.e. when no hidden information determines the label $y_i$. Similarly, learning from noisy labels can be formulated with a single hidden variable $\mathbf{h}_i$, which determines whether the original label remains intact or is corrupted. Knowledge of this hidden information makes the underlying function $f$ deterministic, but $g$ does not have access to the information about the corruption.

## 3 RELATED WORK

The problem formulated in Section 2 is multifaceted and occurs in several different learning paradigms, thus we can only give a brief overview of how it manifests in practice.

Our initial experiment on *MNIST* (Section 4.1) is closely related to research in the area of noisy labels (Snow et al., 2008; Khetan et al., 2018), which extends to crowd-sourcing (Sheng et al., 2008; Karger et al., 2014) and aggregating labels from different labellers. It also shares commonalities with active learning (Settles, 2012), which in our case is a problem of deciding whether to re-sample an existing example to improve label quality or to obtain a new sample to increase overall training data quantity. Importantly, most of this research aims to improve data distribution to the labellers or to reduce bias post-sampling, which differs significantly from choosing a sampling frequency a priori. In addition, they deal with categorical or binary labels, while we mostly address domains with real-valued evaluations.

In the context of imperfect-information games, numerous different approaches exist to, explicitly or implicitly, evaluate an information set. Techniques such as Perfect Information Monte Carlo (Long et al., 2010; Furtak & Buro, 2013) combine evaluations of different perfect-information searches into a policy for the imperfect-information state and Information Set Monte Carlo Tree Search (Whitehouse et al., 2011) operates on information sets. Counterfactual Regret Minimization (Zinkevich et al., 2007), as well as its successors, and ReBeL (Brown et al., 2020) learn the utility of individual information sets through self-play. Recent work by Blüml et al. (2023) samples individual world states and constructs imperfect-information policies based on their evaluations. In essence, most techniques for solving imperfect information games involve estimating the value of information sets, further motivating the importance of the question which we aim to answer.

Finally, this paper is concerned with learning to approximate the value of a set, which is defined as the mean of all of its items, by sampling a subset of it. At its core, this general idea that sampling more states from the information set will lead to a more accurate estimate of its overall value is simply an instance of the law of large numbers and thus finds application in a variety of problems. This trade-off between quantity and quality of evaluations is also analogous to the choice of rollout policy in classical Monte Carlo Tree Search (Browne et al., 2012), where one has to decide between random rollouts (fast, and thus allowing a larger quantity, but less informative) and more sophisticated rollout-policies (slower, thus limiting their number, but better at approximating true behaviour).

## 4 EXPERIMENTS

In this section, we present a series of experiments designed to investigate the trade-off between (a) obtaining a fresh training example and (b) increasing the labelling quality of an existing sample. The learner has a limited budget of total sampling queries $N$ and can decide whether to spend it on

(a) or (b). We construct multiple runs with different numbers of samples $k$ per training example. Each query yields one possible target value, which will be aggregated into an overall label that is used for training. The source code for all experiments will be made public after reviews to preserve anonymity.

## 4.1 MNIST

In the first experiment, we create an imperfect-information adaption of *MNIST* (Deng, 2012). This serves as a first investigation into the effect of sampling different amounts of labels for a single example.

### 4.1.1 SETUP

Each original training example $(\mathbf{x}_i, y_i)$ is annotated with a hidden variable $\mathbf{h}_i = (h_i)$, which is set to $\emptyset$ with probability $(1 - p)$, or, with a probability of $p$, set to a uniform randomly sampled class label. If $h_i = \emptyset$, the original label is used in the transformed dataset $(\mathbf{x}_i, \hat{y}_i)$, i.e., $\hat{y}_i \leftarrow y_i$, otherwise $y_i \leftarrow h_i$ is used. Effectively, this adds uniform class noise to the learning problem, but it is modelled as an imperfect information scenario, where knowledge of $h_i$ would facilitate the learning task, as a perfect-information classifier could learn that values of $h_i \neq \emptyset$ directly determine the class label. However, the learner only has access to the imperfect information $(\mathbf{x_i}, \hat{y}_i)$. It thus has to be able to deal with possibly contradicting samples $(\mathbf{x}_i, \hat{y}_i)$ and $(\mathbf{x}_j, \hat{y}_j)$, where $\mathbf{x}_i = \mathbf{x}_j$ but $\hat{y}_i \neq \hat{y}_j$, and effectively learn to associate the expected value $\mathbb{E}[\hat{y}_i] = y_i$ with each $\mathbf{x}_i$. After drawing repeated samples for $\mathbf{x}_i$, they are aggregated via voting for the most frequently observed class; ties are broken randomly.

We train a basic convolutional neural network on online-generated samples with varying values of $k$, representing the number of labels sampled per training example, and the corruption probability $p$. One difficulty is that when comparing the influence of $k$ and $p$ on the training process of the network, we can either regard the performance as a function of the total number of samples generated, the number of gradient updates taken, or the wall time passed. Without knowledge about the relation of these, every choice introduces some bias into the comparison; when regarding only the total number of samples generated, variants with more samples per training example have fewer opportunities to update the parameters of the network, but when only considering the number of updates, runs with fewer validations have no real chance to perform better as they possess a noisier training signal. Using the wall time introduces hardware biases.

### 4.1.2 RESULTS

The initial findings are summarised in Figure 1, where we compare which choice of $k$ leads to the best peak accuracy over multiple runs, either given a budget of 1 million labels generated or 1 million updates performed. Additional training curves can be found in the Appendix (Figures 9 and 10). When equating for the total quantity of generated labels, sampling a label multiple times leads to worse results in almost all cases. Only for very high noise levels, repeated sampling ($k = 3$) is advisable. Note that there can be no difference in performance for $k = 1$ and $k = 2$ in all settings because labelling a sample twice does not lead to a higher chance of returning the true label (see Lemma A.1). Even when equating for the total number of parameter updates, no difference is found between the peak accuracies, with the extremely high 99% noise setting being the only exception, where overall performance is improved with more samples.

Thus, we conclude that sampling multiple labels leads to worse efficiency than using a simple sample for this experiment. However, this could be attributed to the learner's ability to see the same training example multiple times in different epochs, thus mitigating the downside of only sampling once.

## 4.2 TEXAS HOLD'EM POKER

This experiment aims to use the observations from Section 4.1 in a real-world setting where we have to balance the label accuracy with the number of total training examples seen. Here, the learner aims to estimate the win probability of a given 2-card hand of cards in two-player heads-up poker. This is a direct implementation of the problem outlined in Section 2.

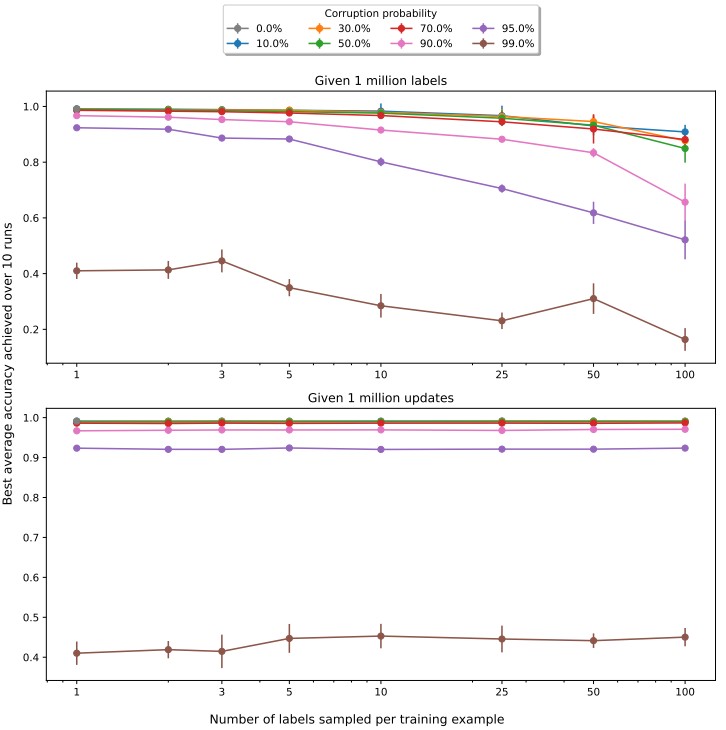

Figure 1: Comparison of best choice (given by highest average test-accuracy achieved) of number of labels sampled per training example for different label corruption levels for a fixed budget of evaluations (top) and a fixed budget of parameter updates of the learner (bottom).

#### 4.2.1 SETUP

In principle, for a given hand $\mathbf{x}$, $g(\mathbf{x})$ could be directly computed as the average over all possible hidden contexts $\mathbf{h_x}$, but doing so would require immense computational resources. Without accounting for symmetry, a player can have $\binom{52}{2} = 1326$ unique Poker hands. One would need to compute all possible arrangements of the remaining cards into two opponent cards and five community cards, i.e. $\binom{52}{2} \cdot \binom{50}{2} \cdot \binom{48}{5} = 2{,}781{,}381{,}002{,}400$ total combinations. For each of these combinations, one needs to evaluate which player won the game and average this for all configurations that pertain to the same player's hand to estimate the overall winning probability of that hand. While public data for the win-chances of a hand exists, such data is only available for the most popular games and computing them is much costlier in other games with higher degrees of uncertainty or more expensive state evaluations. Thus we aim to decrease the computational cost by only sampling parts of the information set instead of enumerating it entirely.

When training the Neural Network, we sample $k$ different configurations of cards for the given hand, evaluate the result of this configuration (0, 1 or 0.5), and train the network to predict the mean of all $k$ samples. Sampling more combinations leads to a smaller difference between the estimate and the ground truth, but more computation is required to generate them, which results in a smaller amount of total hands seen when equating for the total number of evaluations.

#### 4.2.2 RESULTS

As a first estimate, Figure 2 shows the discrepancy between an estimated hand strength through evaluations and the true win chance according to a table [1]. Notable, with only a single sampled configuration, it is impossible to exactly receive the true win chance of most hands as the only possible results are 0, 0.5, and 1, thus resulting in three error clusters of the histogram for one

---

[1]https://www.winallpoker.com/wp-content/uploads/Heads-up-poker-odds-win.pdf

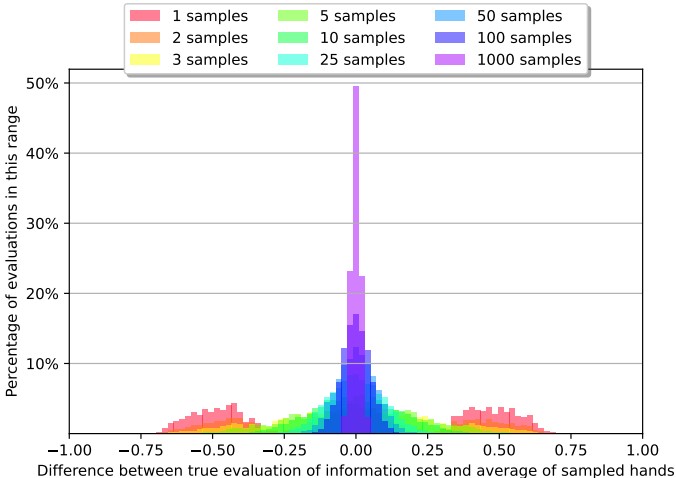

Figure 2: Error in evaluating the change of winning with a given hand pre-flop- in heads-up poker. Estimations are computed by averaging over n samples of possible opponent hands and rivers.

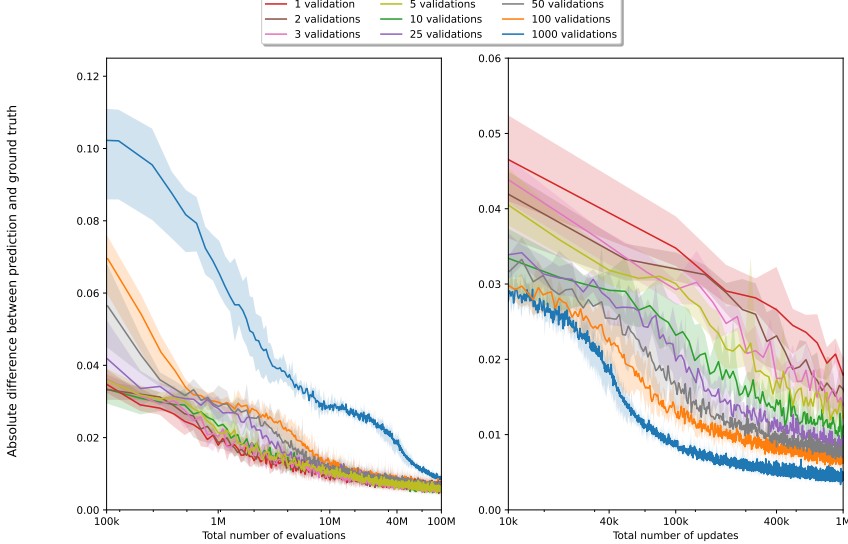

Figure 3: Average training curves of learning to evaluate a poker hand with different numbers of evaluations per training example. The $x$-axis is logarithmically scaled either by the total number of hand evaluations (top) or by the total number of update steps made (bottom).

sample. This means that repeated sampling not only increases the probability of being close to the true evaluation, it also improves how close the sampled evaluations can potentially be.

The training process (Figure 3) shows that training with fewer evaluations per example leads to much quicker progress when regarding the performance in relation to the total number of evaluations, but when the examples have higher-quality evaluations, each update is more meaningful. However, comparing the best versions (Figure 4), we see that even when equating for the total number of evaluations requested, using a single evaluation leads to worse peak results than using two, three, five, and ten sampled evaluations. When equating for training updates, more evaluations perform strictly better than less, which stands in contrast to Section 4.1, where peak results did not improve with more samples in almost all settings. It is not clear what causes this discrepancy, but we speculate that it is related to using real-valued evaluations as opposed to categorical labels, which might be

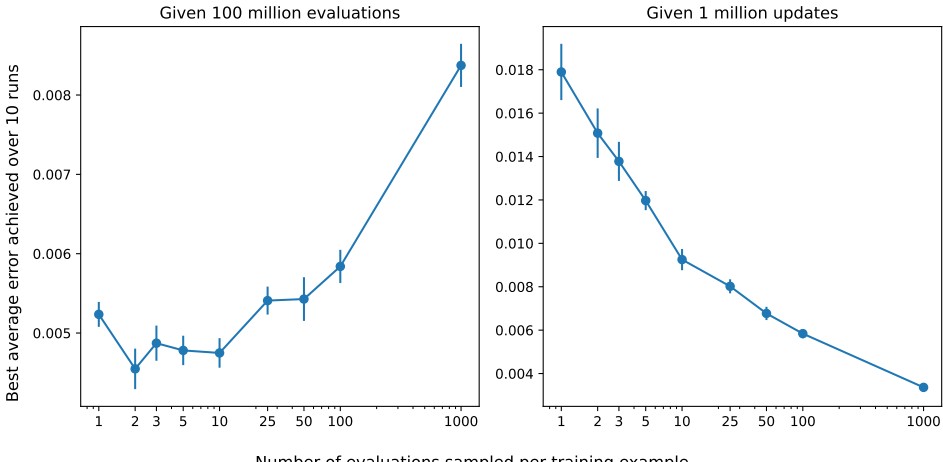

Figure 4: Average lowest received error for the different options of hand validations given a total budget of either 100M hand evaluations or 1M training updates.

more forgiving. All in all, these results suggest that multiple samples are useful for this setting, but naturally, spending too much computation on a single example degrades the overall performance as total training data quantity diminishes.

### 4.3 RECONNAISSANCE BLIND CHESS

Finally, we test one last setting; *Reconnaissance Blind Chess*.[2] *RBC* is an imperfect-information adaption of chess, where players receive limited information about the opponent's moves. When training agents to play this game, it is highly useful to be able to evaluate a specific situation (i.e. the received observations at one point in time), and evaluation functions for regular chess are readily available (e.g., from open-source programs such as Stockfish[3]. Thus, computing the average evaluation of all states in an information set is an intuitive approach, but doing so is infeasible in the game, as the information set can involve thousands of different game states. As such, this game is a real-world example of the problem we are trying to investigate: How can we best invest a given computational budget to generate the most informative training information?

#### 4.3.1 SETUP

For this experiment, training data is created offline in advance for each $k$, thus allowing each Neural Network to train without requesting additional evaluations. Each learner has a fixed budget of 1 million state evaluations, which are calls to a Stockfish engine, that can be arbitrarily distributed among different information sets. Based on the previous results, sensible values of $k$ were chosen as $\{1,2,3,5,10,25,50,100,1000\}$, thus resulting in datasets of approximately $\{1M, 500k, 333k, 200k, 100k, 40k, 20k, 10k, 1k\}$ examples respectively.[4] Importantly, the number of potential public-information states to get evaluated in this experiment is much higher than in the previous Sections 4.1 and 4.2. For *MNIST*, the training data is limited to 60,000 images and in *Poker*, there are 1326 unique 2-card hands. However, in *RBC*, the number of potential observations which form one information set is estimated to be $10^{139}$ (Markowitz et al., 2018), enormously larger than our training datasets, thus minimising the probability of overlapping training example and increasing the importance of meaningful target value estimations.

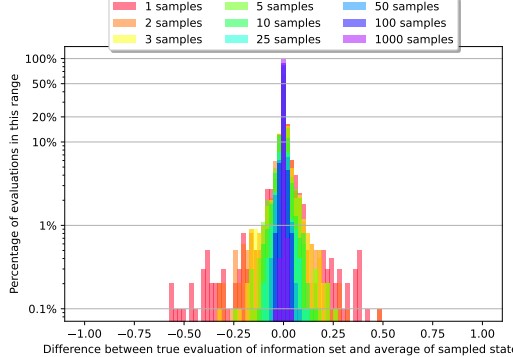

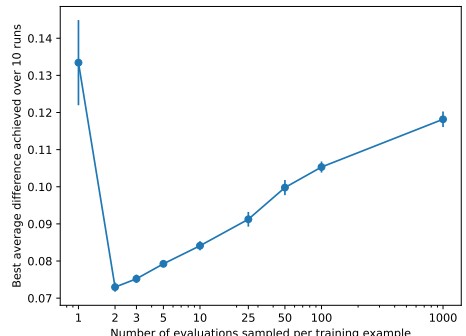

Figure 5: Error in evaluating the odds of winning for a given observation. Estimations are computed by averaging over $k$ samples of possible board states, true evaluation is defined as the average over the whole information set. Note that the y-axis is in a logarithmic scale to improve readability.

Figure 6: Average lowest received error for training datasets created with a total budget of 1M sampled boards. For each training example, $k$ different boards are sampled from the information set, leading to 1M/$k$ different training examples.

### 4.3.2 RESULTS

First, regarding the differences between the true evaluation of an information set and the approximated one given by $k$ samples (Figure 5), we receive the expected result: Sampling only a small number of states can lead to a large discrepancy between the approximation and the ground truth, but we see diminishing returns, such that sampling more than 50 positions only leads to slight improvements in the approximation. Thus, we would expect that sampling multiple states is beneficial, but more than 50 states should lead to meaningful degradation in performance due to the corresponding large reduction of training data quantity.

This observation is confirmed by the results in Figures 6 and 7. The training curves in Figure 7 show that training with a single sample results in poor overall accuracy due to the noisy training signal, but vast oversampling reduces the number of different training examples too much. Figure 6 shows that $k = 1$ and $k = 1000$ are the worst choices in this experiment. Similarly to the findings of Section 4.2, we see that multiple samples are aiding training.

## 5 SUMMARY AND CONCLUSION

With this work, we provided the first experimental results on the influence of sampling different numbers of states from an information set to enable a neural network to learn an evaluation of the whole set. For a given task, a total budget of $N$ evaluations is given, which are distributed among samples from different information sets, varying how many states are obtained from each ($k$). Thus, we investigated the trade-off between the overall number of training samples generated and the accuracy of their associated labels.

As a first observation, the trade-off is additionally influenced by the cost of generating evaluations and the cost of making an update to the learner. Thus, the specific choice of $k$ for one domain will be related to the balance of these costs.

To answer the initial question of how a fixed evaluation budget should be distributed, we find that in the *MNIST* setting (Section 4.1), sampling multiple labels does not lead to better performance in the majority of conditions. This could be a result of the task being rather simple, such that

---

[2]https://rbc.jhuapl.edu/

[3]https://stockfishchess.org/

[4]The exact numbers vary slightly because the information set can consist of fewer states than $k$, thus exhausting it completely.

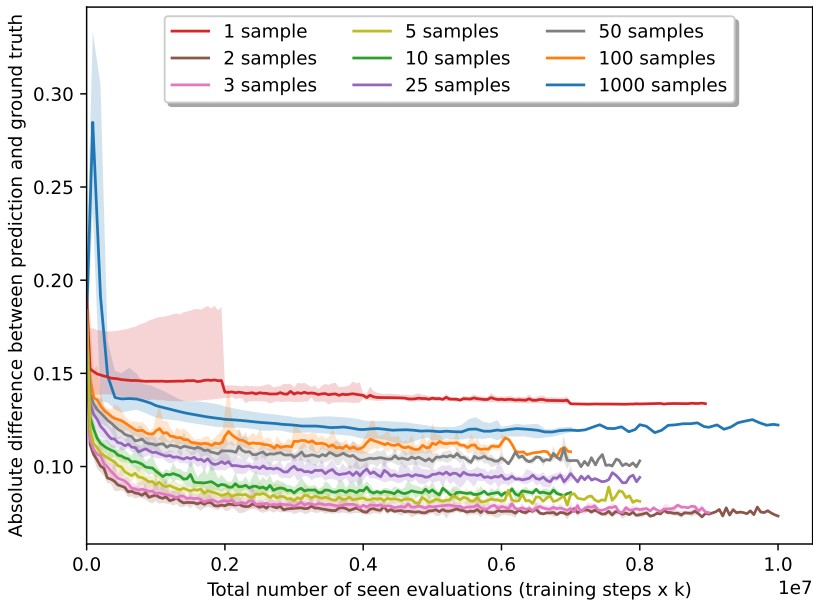

Figure 7: Average training curves of learning to evaluate a history of observations with different numbers of unique board states sampled from the information set per training example. X-axis scaled by the total number of evaluations seen. Curves vary in length because the Neural Network is trained until no further improvements are seen, which happens at different points in time.

label inaccuracy does not significantly impact the performance. However, it may also suggest that when dealing with categorical labels, the benefits of sampling multiples diminish since a single label can already adequately represent the target. Conversely, in the two real-valued tasks (Sections 4.2 and 4.3), we find that generating multiple evaluations consistently improves the performance and efficiency of the neural network. In these two domains, *Heads Up Poker* and *Reconnaissance Blind Chess*, we find that sampling two evaluations per training example lead to overall best results and only using one sample did not perform well compared to the other options. As the results for both domains were similar, we speculate that these findings will translate to more scenarios, but more work is required to validate this.

## 6 FUTURE WORK

We see multiple intriguing lines of further work based on these initial findings. First, we here assumed no agency over the process of sampling from the information sets and no possibility of varying the number of sampled states online. Being able to change either of those assumptions will likely lead to better results and some strategies have previously been outlined by Sheng et al. (2008) for categorical tasks. Secondly, it is unclear whether real-world scenarios exist where very high numbers of samples are applicable. Our first experiment (Section 4.1), albeit artificial, hinted that such settings might exist in niche cases. Finally, while our general formulation holds for other distributions of states, we used a uniform distribution of states for our experiments. While this assumption is sensible for the first two of our experiments, information sets like the ones processed in Section 4.3 do not have uniform distributions in practice. Knowledge of this, or even access to a proxy of such a distribution would lead to more accurate estimations in real-world tasks. Whether a non-uniform distribution changes the best choice of sampled evaluations will be investigated in the future.

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

## A APPENDIX

### A.1 MNIST

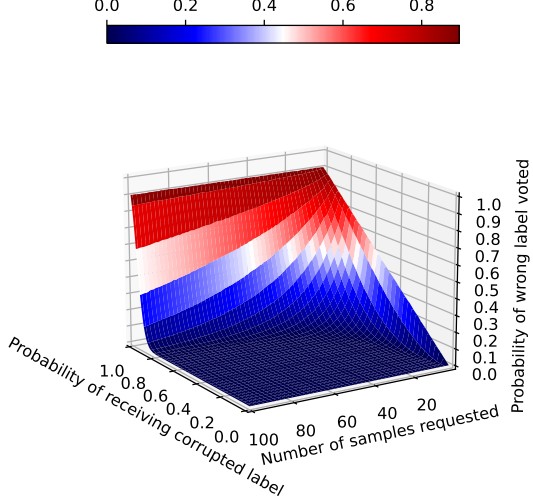

Figure 8: Probability of voting the wrong label for a sample in relation to the number of individual samples requested and the probability that any label was corrupted to a random label. Large areas of the space have a low chance of corruption, so repeated sampling will likely not be needed for those. For the higher-noise settings, repeated sampling of one label might lead to improved training.

**Lemma A.1.** *When majority-voting the label that shall be returned for a given example, sampling exactly $k = 2$ different labels does not improve the probability of returning the true label of that example compared to $k = 1$.*

*Proof.* Let there be $c$ different potential labels for a given example with one label $l_+$ being correct and all labels $l_{-,i}$ being incorrect. $p_+$ denotes the probability of receiving the true label and $p_-$ denotes the probability of receiving each individual wrong label. Naturally, $p_+ + (c - 1) \cdot p_- = 1$. When sampling exactly one label, the probability of receiving $l_+$ is $p_+$ and the probability of receiving any $l_-$ is $1 - p_+$ as given by the definition. When sampling two labels, three different options arise:

1. We receive $l_+$ twice, this occurs with probability $p_+^2$ and will return $l_+$ with probability 1.

2. We receive $l_+$ and any $l_-$ (or vice versa), which occurs with probability $2 \cdot (p_+ \cdot (1 - p_+)) = 2 \cdot (p_+ - p_+^2)$. In this case, a tie between two labels occurs, and a random label of both will be returned, i.e. $l_+$ is returned with probability $1/2 \cdot 2 \cdot (p_+ - p_+^2) = p_+ - p_+^2$.

3. We receive any two $l_-$, which happens with probability $(1 - p_+)^2$ and will return $l_+$ with probability 0.

Thus, we receive $l_+$ with probability $p_+^2 + p_+ - p_+^2 + 0 = p_+$, which is the same probability as when sampling one label. □

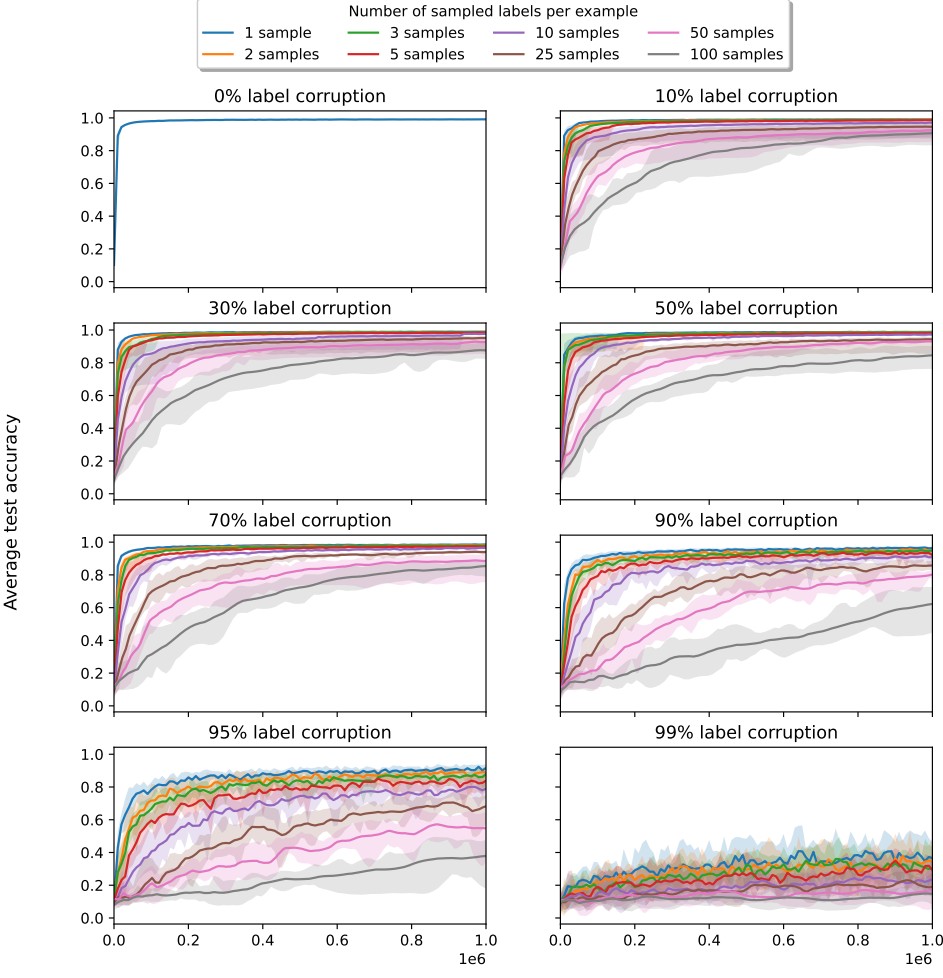

Figure 9: Comparison of test-accuracy over time for different combinations of label corruption rate and number of labels sampled per training example. The X-axis is equalised by the total number of labels that were generated.

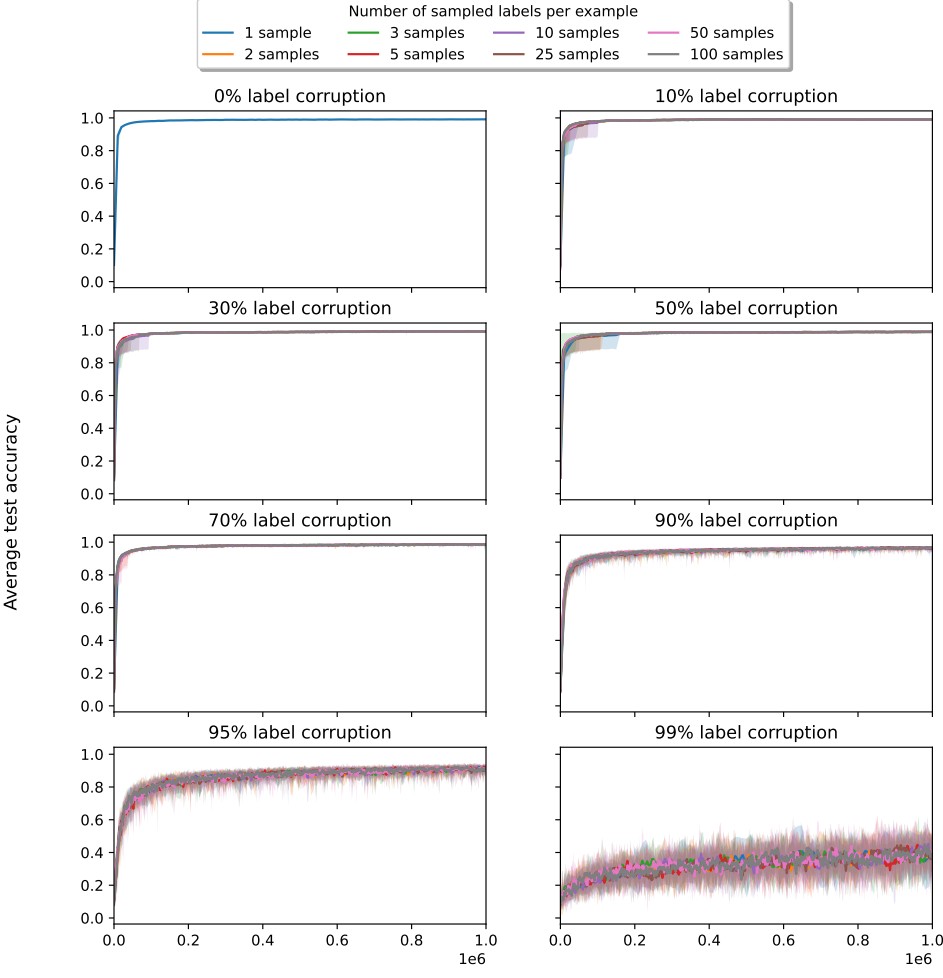

Figure 10: Comparison of test-accuracy over time for different combinations of label corruption rate and number of labels sampled per training example. The X-axis is equalised by the total number of parameter updates performed.

