# OpenReview forum: "On Sampling Information Sets to Learn from Imperfect Information"
_ICLR.cc/2024/Conference — Submitted to ICLR 2024_

### Official Review · Reviewer_ufB6 · 2023-10-12

**Soundness:** 3 good
**Presentation:** 3 good
**Contribution:** 1 poor
**Rating:** 3
**Confidence:** 3

**Summary:**

The paper investigates the question of value estimation of information sets in games from a learning perspective, formulating the question as essentially one of supervised learning with noise and hidden information. Through extensive experiments, the paper evaluates the effect of sampling multiple times from the hidden information.

**Strengths:**

The paper addresses a problem of importance to the game theory community, namely, value estimation of information sets in games. The paper is also clearly written and easy to understand.

**Weaknesses:**

The paper's main problem seems to essentially reduce to supervised learning: namely, given a dataset $(x_i, y_i)\_{i=1}^n$, estimate a function $g(x) := \mathbb E_{y|x} y$. From this perspective, as I will discuss below, the results in the paper are, in my opinion, not very illuminating given what is already well understood about supervised learning. Perhaps most critically, the paper does not seem to make any use whatsoever of the hidden information $h$ in any of the proposed techniques, preferring instead to simply sample from the conditional distribution $y|x$ and use those samples to compute some summary statistic (majority vote, or average value) to feed to the learner instead.

This paper is framed in such a way that the main application is supposed to be in games. But this bothers me: in games, the assumption of the distribution $h|x$ being known is very false, since in practice the opponent has control over incomplete information (e.g., in poker, by choosing its range in any given situation). I would argue that this lack of knowledge of the distribution---not any issue of statistical estimation---is the fundamental barrier to good value estimation in games.

I am also confused about the setup for the MNIST experiment. From what I am understanding, for each training instance $x_i$, you draw $k$ samples $\hat{y}\_i^{(\kappa)}$ ($\kappa=  1, \dots, k$) and then pass the majority vote of these samples, $\hat y_i$, to the learner as the training instance. But this would not optimize the above objective function: indeed, for example, if $p = 0.9$ (say) and $k$ is very large, one would expect $\hat y_i = y_i$ always for every $i$ (simply by LLN) and therefore the classifier would learn to always output the true class. But that would be wrong: when $p = 0.9$, the expected value $g(x) := \mathbb E_{(h, y)|x} y$ is not the true class, due to the high noise. Measuring the resulting classifiers according to their test accuracy also seems wrong, for the same reason (it's not the objective stated by the paper). I think in the MNIST case, like in the game cases, the aggregator should be the average $\mathbb E_\kappa \hat{y}\_i^{(\kappa)}$ (where the $\hat{y}\_i^{(\kappa)}$ are to be interpreted as one-hot vectors), not the majority vote.

It seems also pretty clear to me why the results in Figure 4 are the way they are: with a fixed number of evaluations, having more samples per instance means having fewer unique instances, which could lead to overfitting; on the other hand, with a fixed number of updates, having more samples per instance only means less noise in the training data, which is a good thing.

**Questions:**

In the game setting, how does the lack of knowledge of the hidden information $h$ affect the results? In particular, as suggested above, in games, the distribution of $h$ is generally unknown and may be manipulated by an adversarial opponent, and in fact the critical thing is to be able to play well *regardless* of what distribution the opponent may choose. For example, in poker, it is most certainly not the case that, for a given hand, all 2 trillion possible combinations of opponent hands and community cards are equally likely since (among other things) good players will fold many of their starting hands!

---

### Official Review · Reviewer_p622 · 2023-10-30

**Soundness:** 1 poor
**Presentation:** 1 poor
**Contribution:** 1 poor
**Rating:** 1
**Confidence:** 5

**Summary:**

This paper provides some experimental results on evaluating the information set. The motivation is unclear and I cannot foresee any possible applications. The presentation is poor and I barely understand the point the authors want to argue.

**Strengths:**

None.

**Weaknesses:**

* The unclear motivation: The authors argue that the motivation comes from evaluating the game tree of the imperfect information game. However, for evaluating imperfect information game, one should also take opponents' policy into account and this paper completely ignore this. Hence, I don't think this paper can have any help for imperfect information games. Learning with noisy labels can be an alternative understanding but I don't think the authors show some rigorous consideration on that.
* The poor presentation: I believe statements in Section 2 have no significance -- they are just standard Monte Carlo approximations. Until I go through the experiment I realize the authors may want to argue that how to generate samples that can help the neural network training. I don't think the authors make this clear in previous sections and there should be much more discussion on this before experiments.
* Limited significance: For me I think this is just an experimental report and I cannot get any meaningful conclusion from this paper. The authors make no discussion on the reason to use these methods and what are the potential benefits/drawbacks using these methods in a rigorous manner. I also believe that the proposed problem is a delicate learning theory problem and at least there are lots of work on learning with noisy labels that propose different algorithms with theoretical justifications to achieve strong empirical performances. This paper instead shows no effort on understanding the problem and improving the performance upon existing baselines.

**Questions:**

I would like to ask for the reason that missing a formal methodology part in this paper to discuss the proposed method and the significance compared with the existing work.

---

### Official Review · Reviewer_nuwk · 2023-10-31

**Soundness:** 1 poor
**Presentation:** 1 poor
**Contribution:** 1 poor
**Rating:** 1
**Confidence:** 4

**Summary:**

This paper investigates the impact of sampling of an information set on the approximation of the  corresponding information set value in imperfect-information games. Specifically, it studies how many states from a given information set should be sampled, and how many times should a selected state be sampled. Different experiments are performed, and some observations are obtained.

**Strengths:**

+ Partial observability is a practical issue in real-world decision making problems, and this paper tackles this issue from a perspective based on information set.

**Weaknesses:**

- This paper is not scientifically sound. It does not provide a concrete mathematical problem formulation, and it does not provide a principled approach to address a well defined problem either. The main contribution of this work is some empirical observations, which does not provide much insight or guideline on how to design an efficient sampling strategy to mitigate the partial observability in decision making problems.

**Questions:**

- Is it possible to formulate this as a well-defined problem and provide a more principled approach to address it?

---

### Meta-Review · Area_Chair_PpvJ · 2023-12-06

**Metareview:**

The reviewing team agrees that the paper presents crucial shortcomings that make the paper unsuitable for presentation in this version.

The reviewers criticize lack of soundness (#nuwk, #p622), and poor presentation. Reviewer #ufB6 acknowledges the importance of the problem but criticizes the paper for targeting games under the unrealistic assumption of a known distribution.

**Justification For Why Not Higher Score:**

The lack of soundness and shortcomings in the formalization justify the score.

**Justification For Why Not Lower Score:**

N/A

---

### Decision · Program_Chairs · 2024-01-16

Reject